# The Role of Extracellular Vesicles as Shuttles of RNA and Their Clinical Significance as Biomarkers in Hepatocellular Carcinoma

**DOI:** 10.3390/genes12060902

**Published:** 2021-06-11

**Authors:** Eva Costanzi, Carolina Simioni, Gabriele Varano, Cinzia Brenna, Ilaria Conti, Luca Maria Neri

**Affiliations:** 1Department of Translational Medicine, University of Ferrara, 44121 Ferrara, Italy; eva.costanzi@unife.it (E.C.); gabriele.varano@unife.it (G.V.); cinzia.brenna@unife.it (C.B.); ilaria.conti@unife.it (I.C.); 2Department of Life Sciences and Biotechnology, University of Ferrara, 44121 Ferrara, Italy; carolina.simioni@unife.it; 3Laboratory for Technologies of Advanced Therapies (LTTA)—Electron Microscopy Center, University of Ferrara, 44121 Ferrara, Italy

**Keywords:** extracellular vesicles, RNA, non-coding RNA, liquid biopsy, biomarker, hepatocellular carcinoma

## Abstract

Extracellular vesicles (EVs) have attracted interest as mediators of intercellular communication following the discovery that EVs contain RNA molecules, including non-coding RNA (ncRNA). Growing evidence for the enrichment of peculiar RNA species in specific EV subtypes has been demonstrated. ncRNAs, transferred from donor cells to recipient cells, confer to EVs the feature to regulate the expression of genes involved in differentiation, proliferation, apoptosis, and other biological processes. These multiple actions require accuracy in the isolation of RNA content from EVs and the methodologies used play a relevant role. In liver, EVs play a crucial role in regulating cell–cell communications and several pathophysiological events in the heterogeneous liver class of cells via horizontal transfer of their cargo. This review aims to discuss the rising role of EVs and their ncRNAs content in regulating specific aspects of hepatocellular carcinoma development, including tumorigenesis, angiogenesis, and tumor metastasis. We analyze the progress in EV-ncRNAs’ potential clinical applications as important diagnostic and prognostic biomarkers for liver conditions.

## 1. Extracellular Vesicles

### 1.1. Definition and Classification

Extracellular vesicles (EVs)are defined as lipid bilayer particles naturally released from cells into the extracellular space. They became attractive in the research field when their potential role in cellular crosstalk was discovered [1].

Several studies highlighted that EVs mediate cell-to-cell communication in various biological processes, recognizing them as an additional class of signal mediators, such as cell-to-cell direct interaction or secretion of soluble molecules, i.e., growth factors, cytokines, metabolites, and hormones [2,3,4]. This intercellular communication mechanism allows the delivery of a particular cargo of messages to EV-accepting cells.

This functional cargo varies according to the cell type of origin and the physiological or pathological status of cells when they package and secrete EVs.

EVs are carriers of different molecules [5], such as proteins [6], bioactive lipids [7], and nucleic acids [8].

General criteria classify EV subpopulations based on their biogenesis, adding other hallmarks such as density, size, shape, internal content, surface molecules, and cellular origin [9]. 

Based on their origin, EVs are classified into two main types: exosomes (Exs) and microvesicles (MVs). EVs originating from an intracellular endocytic trafficking pathway are called “exosomes”, whereas EVs, which are formed directly by outward budding of the plasma membrane (PM), are defined as “microvesicles”, “ectosomes”, and “microparticles”. Given their nature, Exs have a typically rounded morphology with variable diameter from 50 to 150 nm and a buoyant density of 1.10–1.14 g/mL; in contrast, MVs appear more heterogeneous in shape and size with a diameter that varies from 50 to 500 (up to 1000) nm (Figure 1) [10]. 

Newer techniques (such as cryo-TEM) led to the finding that exosomes’ previous ‘‘cup-shaped’’ morphology was an artefact related to fixation for TEM analysis. When observed in a close-to-native state by cryo-electron microscopy (cryo-EM), Exs have a rounded shape [10,11].

Since in some isolation approaches, the nature of EVs according to their biogenesis has not been found, the recommendations of MISEV 2018 (i.e., the main consensus of the largest group of EV experts) have recently proposed referring to EV biophysical characteristics such as their size (“small EVs” (sEVs), “medium/large EVs” (mEVs), and “large EVs” (lEVs)), density (light, medium, or heavy), or their biochemical composition, such as the co-presence of protein markers CD81+/CD9+/CD63+ in EVs [12,13].

In this review, the term “EVs”, including both Exs and MVs, has been used since it is not easy to ascertain EV subtypes using the current purification methods often obtained in mixtures of heterogeneous vesicle subsets [14].

### 1.2. Biological and Functional Features of EVs

EVs are involved in a plethora of biological processes such as inflammation, immune response, neurological diseases, and cancer. 

A study on Exs-mediated activation of T cells revealed a mechanism in which the Treg cells transferred Exs-associated miRNAs to other immune cells, including T-helper 1 cells, with suppression of proliferation and cytokine secretion [15]. In adipose tissue, M1 macrophages released Exs containing miRNA-155 that targeted adipocytes, suppressing the expression of peroxisome proliferator-activated receptor-γ (PPAR-γ), that makes them insulin resistant; in contrast, M2 macrophages secreted Exs containing miR-223 that, renders them sensitive to insulin [16].

Cancer-derived EVs influence both stromal [17] and tumor cells in each phase of cancer development. EVs can induce endothelial proliferation and neovascularization as demonstrated by an in vitro study showing that glioblastoma-derived EVs increase levels of vascular endothelial growth factor (VEGF) in endothelial cells and activate VEGF receptor 2 in an autocrine manner [18].

Moreover, EVs can enhance tumor cell migration and invasion: Wnt11-loaded CD81-positive EVs can induce protrusive behavior of breast cancer cells, migration in vitro, and metastases in vivo [19].

One of the most relevant features of EVs is that, as previously demonstrated, EVs are secreted from almost all cell types and can be retrieved in a wide variety of human body fluids and secretions, such as blood (or serum/plasma), urine, breast milk, saliva, synovial fluid, amniotic fluid, cerebrospinal fluid, ascites, and bronchoalveolar lavage fluids [20].

From these different sources, EVs can be isolated, purified, and then biochemically and functionally characterized [21].

Thanks to their ubiquity and self-replenishing efficiency, EVs are considered reliable biomarkers for diagnostic and disease-monitoring purposes [22].

Another aspect that makes them an ideal biomarker is that the content of these vesicles reflects the pathological state of the cells and tissues of origin, thus representing a valuable tool for monitoring the onset, progression, and prognosis of the disease, as well as providing a system to analyze therapeutic efficacy [23].

Indeed, it has been shown that EVs released by tumor cells during malignant progression contain proteins [24], nucleic acids [25,26], and lipids [27] that can be used as markers of the neoplastic and metastatic phenotype. An example is the identification of caveolin-1, a protein associated with the metastatic behavior of tumors, in plasma EVs. The detection of caveolin-1-associated EVs in plasma can, therefore, be considered a useful tumor biomarker [28]. 

Cancer-derived EVs can be isolated using membrane-specific proteins from cancer tissues. For example, the presence of fibronectin on the membrane of circulating EVs was revealed as a tumor-specific antigen, to detect breast cancer [29]. In a recent publication reporting the analysis of lipids in EVs isolated from plasma of 20 pancreatic cancer patients and healthy controls, it was revealed that specific lipids, LysoPC 22:0, PC (P-14:0/22:2), and PE (16:0/18:1), are correlated with tumor stage, and, further, PE (16:0/18:1) was associated with a patient’s overall survival [30].

Furthermore, the importance of EVs lies in their ability to influence the phenotype and functions of cells either nearby or distant from the producing cells, modulating their activities also toward the development of pathophysiological conditions. This is possible due to the presence in EVs of nucleic acids [31].

This review summarizes the current knowledge of EV RNA content, with a specific focus on their ability of mediating the communication between normal and pathological liver cells.

## 2. Extracellular Vesicles as Carriers of RNA Molecules 

### 2.1. Classes of RNA Molecules Retrieved in EVs and Their Biological Function 

The discovery that EVs can carry nucleic acids revealed their crucial role in horizontal genetic transfer [31,32].

The circulating RNAs associated with EVs can reach cells other than the originating, both in neighboring cells and in cells located elsewhere in the body, and, once inside, can influence gene expression.

EVs can contain messenger RNAs (mRNAs) [33] and non-coding RNAs (ncRNAs) of different length, including long non-coding RNAs (lncRNAs) [34], microRNAs (miRNAs), and circular RNAs (circRNAs) [35].

Valadi and colleagues (2007) carried out the first study in which the presence of mRNA was investigated in EVs derived from a mouse (MC/9) and human mast-cell lines (HMC-1) and primary bone marrow-derived mouse mast cells (BMMC) [31].

LncRNAs are RNA molecules characterized by a nucleotide size > 200 bp and lack of protein-coding sequences [36]. In 2014, Gezer et al. identified lncRNAs, including MALAT1, HOTAIR, lincRNAp21, GAS5 (growth arrest-specific 5), TUG1 (taurine upregulated gene 1), and ncRNA-CCND1 (cyclin D1) in EVs derived from HeLa and MCF-7 cell lines. The identified expression patterns of lncRNAs were different in EVs compared with their parental cells [37].

miRNAs are small ncRNA molecules with a length of approximately 18–24 nucleotides, which pair with a specific sequence of mRNAs with imperfect binding [38].

Therefore, they can regulate hundreds of transcripts, reducing and/or increasing mRNA degradation, removing specific resident proteins in the cells [39]. miRNA expression has a tissue-specific profile pattern, and their expression is impaired in many diseases, including cancer [40,41,42].

Finally, circRNAs are circular RNA molecules composed of a covalently closed loop structure, lacking a poly-A tail or 5′ to 3′ polarity [35,43]. Growing evidence has shown the presence of circRNAs enriched in EVs, involving various biological processes of cancer, particularly malignant tumor metastasis [44]. A recent study showed that circRNAs increased their level two-fold in EVs released from MHCC-LM3 liver cancer cells when compared to their parental cells [35]. Indeed, this research group showed that the sorting mechanism of circRNAs is a process linked to the regulation of related miRNA levels in parental cells [45].

The circRNA expression profiles were investigated in EVs released from three isogenic colon cancer cell lines diverging for KRAS mutation compared to their parental cells. Dou and colleagues identified that the circRNA levels are higher in EVs than their parental colon–rectal cancer cells [46]. 

The subcellular localization of RNAs and the EV subtypes positively influences their loading.

Interestingly, a selective sorting process was identified for specific RNAs, sharing a short sequence called hEXO motif during the hepatocytes’ EV formation. The main component of this loading machinery belongs to the synaptotagmin-binding, cytoplasmic RNA-interacting protein (SYNCRIP) complex, which directly binds to some miRNA enriched in EVs [47].

Different studies demonstrated that EVs produced by different cell types presented different RNA content [48], depending on EV subcellular source and cell physio-pathological conditions [49]. An increasing amount of RNA molecules have been found to be aberrantly expressed in human cancers [50].

The up- or downregulation of specific ncRNAs, associated with disruption of cells’ physiological mechanisms, may lead to diseases [51]. Alterations in the action of miRNAs or their biogenesis processes can be used to indicate disease prognosis in a patient.

For example, alterations in miR-122 and miR-33 levels are linked to the development of obesity, hepatic steatosis, and hepatocellular carcinoma (HCC) [37]. miR-33a is an important regulator of cell proliferation and apoptosis by acting on PPARα (peroxisome proliferator-activated receptor α). Functional experiments of miR-33a gain- and loss-of-function demonstrated that its overexpression in Huh7 hepatocarcinoma cells triggers increased proliferation by reducing PPARα levels. In contrast, its inhibition in HepG2 hepatocarcinoma cells reduced cell proliferation and induced apoptosis due to the hyperactivation of PPARα expression. miR-33a is considered a potential prognosis marker for hepatocarcinoma patients; high levels of miR-33a correlate with a shorter survival of 5 years [52].

Based on this information, it is clear that ncRNAs’ biological message is related to tumor cell spreading and oncogenic onset. On the other hand, it is not easy to identify a single ncRNA as a specific disease marker because they act with the principle of cooperation. For example, a single gene targets several miRNAs, just as the same miRNA can act on several genes [53]. Thus, it is more likely to identify a pattern of ncRNAs whose expression is related to a specific alteration.

### 2.2. Methodological Approaches to Study RNA Molecules Carried by EVs

RNA molecules can be isolated from biological samples (i.e., cell culture medium or blood/plasma) in two ways: RNAs can be obtained by extracting the total RNA from both EV-associated RNA or free and protein-bound RNA. Alternatively, more accurately, EVs can be isolated from biological samples using a differential centrifugation approach, ultracentrifugation, or other methods, such as size exclusion chromatography, and only EV RNA can be isolated [54].

Currently, there is no gold standard technique for EV isolation and, thus, the method should be chosen based on both the type and amount of EVs.

Conventional methodologies for EV isolation suffer from limitations in separation technology. In particular, the detection of EVs is vulnerable to artefacts partly induced by sample collection and the huge heterogeneity of EV populations. Furthermore, the main approaches are based on EVs’ physical properties (density, solubility, or size), and are not able to separate the tumor-derived EVs from total EVs [55]. 

To overcome this issue, researchers are committed to exploring different immunoaffinity-based approaches to purify tumor-derived EVs. In particular, Sun and colleagues (2020) developed an EV purification system using a multiple marker cocktail to recognize, enrich, and recover HCC EVs secreted from highly heterogeneous HCC [56]. Several biotechnology companies are currently working to develop a quick and easy assay based on precipitation to isolate EVs. These kits often require polyethylene glycol l (PEG1) solutions, that once mixed with samples allow EVs to precipitate at low speed. Precipitation-based isolation is inexpensive, requires no special equipment, and is compatible with both low- and high-sample volumes. Nevertheless, this method suffers from co-isolation of non-EV particles and protein complexes and must be further improved [57].

The analysis of RNA molecules is allowed by different approaches, which include microarrays, quantitative real-time polymerase chain reaction (qRT-PCR), digital PCR (dPCR), NanoString’s nCounter technology, and next-generation sequencing (NGS) [58,59,60]. The main difference between these methods is the sensitivity of the RNA transcript detection.

The most common RNA detection is the microarray analysis because it can detect simultaneously different nucleic acids and can be customized [61,62].

Digital PCR (dPCR) can be considered an alternative to the qPCR approach and provides more accurate data of the nucleic acid target molecule without a standard curve and dependence on amplification efficiency. The hypersensitivity of dPCR allows detecting RNA molecule targets of low abundance below the qPCR’s sensitivity limit [63,64]. This system can easily reveal and quantify the low amount, like EV content, but it is a long procedure and relatively expensive.

The Nanostring platform is a very recent technology to measure RNA expression [65]. The system, formed on a multiplexed probe library, contains two types of probes (capture probe and reporter probe) specific for each nucleic acid molecule to detect. The capture probes are tagged with biotin at the 3′-end, whereas the reporter probes carry a barcode signal at the 5’-end [65]. The probes are mixed with the total RNA, and the hybridized complexes are then immobilized, and digitally detected, thus assessing the level of expression. This technology works without amplification or reverse transcription. Small RNA amounts could be precisely analyzed, and several hundred unique transcripts could be counted in the same reaction because the counts are measured digitally [66]. In a recent study, the RNA content from EVs was analyzed by the nCounter platform, demonstrating this method’s efficacy to detect plasma EV mRNA transcripts [67].

Finally, it was demonstrated that the Nanostring nCounter is a more accurate system than microarrays and comparable in susceptibility to real-time PCR [66]. 

NGS consists of sequencing technology and is supported by different platforms. It offers the advantage to generate a huge amount of sequence data sets, ranging from megabases to gigabases [68].

## 3. Extracellular Vesicle-Derived RNAs Correlated with Hepatocellular Carcinoma

A body of evidence highlights the growing interest in the investigation of EV involvement in liver cancer.

Hepatocellular carcinoma (HCC) is the most common form of primary liver cancer, which generally arises as a direct progression and evolution of chronic liver diseases (CLD), including liver cirrhosis (LC), and is an overly aggressive carcinoma with a poor prognosis [69,70].

EVs can provide a consistent form of the liver intercellular network between hepatocytes, intrahepatic cholangiocytes, Kupfer cells which are liver-resident macrophages, hepatic stellate cells (HSCs), endothelial cells, fibroblasts, infiltrating immune cells, and recruited mesenchymal stem cells (MSCs), given the multicellular nature of liver [71].

HCC-derived EVs can mediate cell growth, modulation of epithelial–mesenchymal transition, migration, invasion of HCC cells, and the angiogenesis process. EVs shuttle biologically active RNAs that may alter the tumor microenvironment, resulting in HCC progression and metastasis [72].

### 3.1. Role of EV-Derived RNA in the HCC Microenvironment Influencing Tumor Progression, Metastasis, and Angiogenesis

The analysis of tumor-associated RNA within EVs could allow the identification of novel biomarker candidates.

Many studies have shown aberrantly expressed tumor-associated protein-coding mRNAs and the expression of specific non-coding RNAs, including miRNAs and lncRNAs selectively enriched in EVs released from different HCC cell lines [73].

EVs derived from HKCI-C3, HKCI-8, and MHCC97L cell lines increased RNAs of lengths ranging between 500 and 4000 nucleotides, compared to their parental cells. The RNA analysis identified mainly mRNA and lncRNA molecules. A limited quantity of ribosomal RNA (18S and 28S rRNA) and mRNA can be translated into proteins in the recipient cell [73].

In Hep3B-derived-EVs, 11 miRNAs (miR-133b, miR-142-5p miR-215, miR-367, miR-376, miR-378, miR-451 miR-517c, miR-518d, miR-520f, and miR-584) were the only ones detected. Likewise, the expression of 20 miRNAs was explicitly discovered in PLC/PRF/5- derived EVs [74].

The results obtained in both cell lines suggest a mechanism driving the hepatic tumor cells to sort a specific set of miRNAs into HCC-derived EVs.

In an exciting study, HCC cells were treated with a neutral sphingomyelinase 2 (nSMase) inhibitor (GW4869) and the expression of miR-16, a miRNA expressed in both originating cells and small EVs, was evaluated. The intracellular expression of miR-16 was unchanged, whereas the extracellular expression of miR-16 in small EVs decreased after incubation with GW4869, compared to controls. This result demonstrates that specific miRNAs from HCC cells could be released into EVs in a ceramide-dependent manner [75].

Recent findings highlighted that some miRNAs, called oncogenic miRNAs or oncomiRs (e.g., miR-21), are able to activate the cell proliferation and inhibit the apoptotic processes, thus regulating HCC growth and development [76].

The main oncomiR, which is highly expressed in almost all solid cancers, including HCC, is miR-21, which is also enriched in tumor-derived EVs [77,78,79,80]. Generally, miR-21 has an anti-apoptotic, pro-survival function in tumor cells. The analysis of miRNAs in HCC-derived EVs showed that miR-21 expression level in EVs was positively associated with the intracellular one in cells and negatively associated with its target genes PTEN, PTENp1, and TETs. Therefore, the EV-miR-21 might modulate the expression of the tumor suppressor genes PTEN and PTENp1, affecting HCC cells’ growth [81].

One target of the EV-derived miRNA was identified in the transforming growth factor-β activated kinase-1 (TAK1) pathway in HCC cells [74].

TAK1 is a kinase involved in hepatic cellular homeostasis and liver pathology, including HCC tumorigenesis [82,83]. Both cytokines and stress stimuli, such as transforming growth factor-β (TGF-β), tumor necrosis factor-α (TNF-α), and interleukin (IL)-1β, can trigger TAK1 [84].

Some evidence demonstrated that HCC development and metastasis are triggered when the suppression of the constitutive expression of TAK1 is induced in hepatocytes [85].

TAK1 modulation can be mediated by EV-derived miRNAs in recipient cells, enhancing tumoral cell growth. Hep3B-derived EVs were incubated with Hep3B cells, and after 24 h, the cell viability and apoptosis were analyzed. The content of EVs derived from Hep3B cells induced a decrease in cell viability of recipient cells and an increase in caspase-3/7. Additionally, it was able to enhance the anchorage-independent growth of tumor cells [74]. Hence, it was clear that the EVs with their cargo have a potent effect on tumoral cell behavior.

EV-derived miR-210 released by hepatic tumor cells is transferred into endothelial cells and leads to tumor angiogenesis, inhibiting SMAD4 (mothers against decapentaplegic homolog 4) and signal transducer and activator of transcription 6 (STAT6) expression [86].

EV-derived miR-103 released by HCC cells enhanced vascular permeability and promoted tumor metastasis by directly targeting endothelial junction proteins, including VE-cadherin (VE-Cad), p120-catenin (p120), and zonula occludens 1 (ZO-1) [87].

EV-derived miR-1247-3p released by highly metastatic HCC cells triggered the activation of β1-integrin–NF-κB signaling in fibroblasts and induced cancer-associated fibroblast activation, promoting tumor metastasis [88].

The selective enrichment of lncRNA in HCC-derived EVs has been demonstrated. TUC339, linc-RNA-RoR (long intergenic non-protein-coding RNA, regulator of reprogramming) and linc-RNA-VLDLR (very low-density lipoprotein receptor) are the prominent lncRNA family members detected and implicated in tumor cell behavior [74,89,90].

EVs isolated from HepG2 cells contain a higher amount of one of the ultraconserved lncRNAs, named TUC339, compared to EVs from non-malignant hepatocytes [91].

The same result has been obtained in EVs isolated from PLC/PRF/5 cells, demonstrating that the most highly expressed lncRNA was TUC339 in HCC cell-derived EVs [92].

The inhibition of TUC339 with short interfering RNA (siRNA) decreased the proliferation and adhesion ability of hepatic tumor cells. Accordingly, the delivery of lncRNA-TUC339 via EVs can be considered a novel signaling mechanism for developing HCC growth and metastasis [74].

Healthy hepatocytes display a low level of lncRNAs, including the long intergenic noncoding RNA, regulator of reprogramming (linc-RoR) [89], which can prevent tumorigenesis and cell proliferation by directly regulating the stability of the c-Myc mRNA [93]. EV-derived linc-RoR released by HCC cells was highly expressed during hypoxia conditions. The increase in EV-derived linc-RoR level in HCC cells decreased the expression of miR-145, a linc-RoR target, resulting in an increase in hypoxia-inducible factor-1α (HIF-1α) and pyruvate dehydrogenase kinase isozyme 1 (PDK1) protein expression [89].

Hence, the EV-derived linc-RoR could promote HCC progression by increasing HCC resistance against adverse environmental conditions, including hypoxia [89].

Most recently, HCC-derived circRNAs were found to display an aberrant expression associated with tumoral characteristics and recent studies reported circRNAs enrichment in EVs released from HCC cells [94].

circPTGR1 is a circRNA with three isoforms enriched in EVs isolated by HCC cell-lines, and its expression level was correlated with tumor differentiation stage, indicating its prognostic potential in HCC patients [95]. The study included the analysis of circRNA expression of EVs derived from three different HCC cell lines: non-metastatic (HepG2), low-metastatic (97L), and high-metastatic (LM3) cells. EVs derived from LM3 cells and containing circPTGR1 enhanced the cell migration and invasion attitude of HepG2 and 97 L cells and, on the other hand, knockdown of circPTGR1 expression in LM3 cells inhibited the migration and invasion of HepG2 and 97L cells [95]. Therefore, circPTGR1 was highly abundant and aberrantly expressed in malignant cells and in cells from patients with metastases, thus showing its contribution to HCC progression and metastasis. It could represent a prognostic biomarker and therapeutic target in HCC (Table 1).

### 3.2. Role of EV-Derived RNA as HCC Suppressors

A large class of miRNAs acts as tumor suppressors, such as miR-122, indicating that HCC-derived EVs are a system that allows the modulation of HCC growth and progress [96]. MiR-122 is a liver-specific anti-proliferative miRNA and is involved in regulating fatty acid and cholesterol pathway as well as normal cell homeostasis and growth, to maintain tumor growth under control [97]. The hepatic decrease in miR-122 expression level could favor the development of steatohepatitis, such as nonalcoholic fatty liver disease (NAFLD) [98]. Studies in NAFLD animal models highlighted the increase in circulating EV-associated miR-122 [98,99,100].

The delivery of EV-miR-122 from normal hepatocytes suppressed tumor progression. However, this effect is inhibited when tumor-initiating cells (T-ICs) start to secrete insulin-like growth factor 1 (IGF-1), arresting miR-122 release from neighboring healthy hepatocytes, thus resulting in a reduction in its anti-proliferative activity and in hepatic tumor development and metastasis [101]. Thus, the expression of miR-122 correlated in the early NAFLD progression with HCC development [102].

Vps4A (vacuole protein sorting 4), a member of the AAA-ATPases (ATPases associated with a variety of cellular activities), was recognized as a tumor suppressor in HCC cells by regulating the release and uptake of EV-derived microRNAs [103].

Wei and colleagues showed that Vps4A inhibited EV function by selectively packaging oncogenic miR-27b-3p and miR-92a-3p into EVs and accumulating tumor-suppressive miR-193a-3p, miR-320a, and miR-132-3p in HCC cells [103]. 

Furthermore, they found that Vps4A reduced the recipient HCC cell response to EVs via selective uptake of exosomal tumor-suppressive miR-122-5p, miR-33a-5p, miR-34a-5p, miR-193a-3p, miR-16-5p, and miR-29b-3p [103].

## 4. Extracellular Vesicle-Derived RNAs as Potential Biomarkers in HCC 

The need to develop accurate and reliable early diagnostic tools to complement and potentially replace invasive liver biopsy, to perform disease stratification and response monitoring for therapeutic interventions, is increasingly growing [104].

New biomarkers’ discovery in early diagnosis and prognosis requires different processes, starting from basic research and validation to clinical implementation.

The final aim is to create clinically available biomarker tests to guide clinical decision-making and improve patient outcomes (Figure 2).

Although liver biopsy remains the “gold standard” method, with limitations either in the absence or presence of cirrhosis [69], “liquid biopsy” is proposed as a novel tool for monitoring HCC development and progression. The liquid biopsy is based on the analysis of different biomarkers, including circulating tumor cells (CTCs), circulating proteins, or cell-free nucleic acids (such as cfDNA) and EVs.

Circulating EVs containing nucleic acids (DNA and RNA) have several advantages as disease biomarkers [105,106].

In plasma/serum, the number of EVs is considerably higher than CTCs, EVs are more stable in the bloodstream, and their cargo is well-protected within the double-leaflet membrane concerning the cfDNAs.

Considering the EV content, they contain various pieces of information, and they cannot be considered as single biomarkers but as a heterogeneous complex of potential biomarkers.

Several studies have focused on EV-associated RNAs as potential diagnostic or prognostic biomarkers [107,108,109].

Most of the studies considered serum as the primary liquid biopsy source for HCC and only one study employed urine as a potential EV source associated with HCC diagnosis.

### 4.1. EV-Derived miRNAs as Prognostic and Diagnostic Markers for HCC

**miR-93.** Considering the important role of EV-derived miRNA in HCC tumorigenesis and that miR-93 increases cancer cell growth via modulating PTEN in several types of cancer [110,111], Xue and colleagues (2018) investigated the role of EV-derived miR-93 as a new diagnostic and prognostic biomarker in HCC [112].

In EVs of tumor sera, they found high expression of miR-93 and its expression level in HCC patients can be considered a diagnostic marker in association with tumor size and TNM (tumor, node, metastasis) stage. Further, the upregulation of EV-derived miR-93 predicts poor prognosis for patients with HCC [112]. 

**miR-21**. Wang et al. (2014) analyzed the miRNA profile from serum EVs in HCC, identifying miR-21 as a candidate biomarker to discriminate patients with liver cancer from those with chronic hepatitis B (CHB) or healthy subjects [78]. In fact, the expression level of serum EV-derived miR-21 was significantly higher in patients with HCC than those with CHB or healthy volunteers [78]. 

Using gene expression arrays to profile miRNAs in tumor and healthy tissue and EVs from HCC patients has been performed. It showed the upregulation of miR-21 in tumor tissue and the plasma EVs with a positive correlation between serum EVs and HCC tissue miR-21 expression, suggesting that miR-21 moved from tissue to bloodstream via EVs [113]. 

Tian and co-authors (2019) showed that the overexpression of miR-21 and miR-10b in EVs from HCC is induced by the acidic microenvironment in HCC, demonstrating that this oncogenic event promotes the proliferation and metastasis of HCC cells. Further, their finding indicated that EV-derived miR-21 and miR-10b can be used as prognostic molecular markers and therapeutic targets of HCC [114].

Another study highlighted that EV-associated miR-18a, miR-221, miR-222, and miR-224 were significantly higher in the serum of patients with HCC than in those with CHB or LC. Serum levels of EV-associated miR-101, miR-106b, miR-122, and miR-195 were lower in HCC than in CHB, whereas there was no apparent difference in miR-21 and miR-93 levels between the three groups [115]. 

A comparative analysis of miRNA level expression between serum-circulating miRNA and EV-associated miRNA in each patient group was also performed in this study. The results show a high correlation of serum-circulating miRNAs and serum EV-associated miR-221, miR-222, and miR-224 in the HCC and LC groups; however, the differences detected in serum miRNA levels were lower than those detected inside the EVs. The data obtained highlight an interesting aspect regarding the possibility of better-discriminating HCC from CHB or LC using serum EV-associated miRNAs when compared to serum circulating miRNAs [115]. 

**miR-9-3p.** The study of Tang and colleagues (2018) investigated the diagnostic use of EV-derived miR-9-3p in HCC, showing lower serum levels of EV-derived miR-9-3p in HCC patients than in healthy donors. Further, the overexpression of EV-derived miR-9-3p reduced the viability and proliferation of HCC cells and additionally reduced ERK1/2 expression, suggesting a potential mechanism for miR-9-3p action [116].

**miR-224.** The expression level of serum EV-derived miR-224 was increased in HCC patients compared to healthy controls, as determined by qPCR, thus suggesting its ability to stimulate the proliferation and invasion of HCC cells [117]. 

EV-derived miR-224 was tested as a biomarker to distinguish HCC patients from healthy controls. The expression of serum EV-derived miR-224 was higher in patients with larger tumors and advanced stages. The correlation between the expression of EV-derived miR-224 and the overall survival of patients has been analyzed. The results demonstrate that the higher the expression level of serum EV-derived miR-224, the shorter the patient’s overall survival, suggesting serum EV-derived miR-224 as a prognostic factor in HCC patients [117].

Currently, detection of early-stage HCC using a serum marker in patients at high risk of developing HCC is challenging. To improve the prognosis of patients with HCC, a reliable serum biomarker for diagnosing early-stage HCC is fundamental [118]. 

In a recent study, the analysis of serum samples from 28 healthy individuals, 60 CLD patients, and 90 HCC patients was carried out. It was found that serum EV-derived miR-10b-5p displayed a relevant diagnostic efficiency in detecting early-stage HCC and serum EV-derived miR-215-5p was recognized as a biomarker for predicting prognosis in HCC patients [118]. In this study, the expression of the two serum EV-derived miRNAs was much higher than that of the circulating serum miR form.

**miR-429.** Li and colleagues (2015) reported that miR-429 is overexpressed in HCC tissue and primary liver tumor-initiating cells (T-ICs). This overexpression is paralleled with HCC-derived circulating EVs and, therefore, may be proposed as a prognostic factor in HCC patients [119]. The high amount and the internalization ability of miR-429 in HCC cells, specifically in epithelial cell adhesion molecule (EPCAM) + T-ICs, contribute to promoting and developing tumor features such as self-renewal, tumorigenicity, malignant proliferation, chemoresistance, and progression [119]. These EVs shuttling and spreading miR-429 into their surrounded target cells induced a novel functional axis with Rb-binding protein 4 (RBBP4) and the transcriptional function of E2F1, POU class 5 homeobox 1 (POU5F1) expression [119].

In particular, miR-429 is able to enhance the transcriptional activity of E2F1 by directly targeting RBBP4, a known tumor suppressor protein, which is downregulated. 

The molecular mechanism regulating miR-429 expression is an epigenetic event involving four abnormal hypomethylated sites upstream of the miR-200b/miR-200a/miR-429 cluster. The EV-derived miR-429 could potentially inactivate T-ICs, thus providing a novel strategy for HCC prevention and treatment [119].

**miR-125b.** Liu et al. (2017) identified the EV-associated miR-125b as a useful prognostic marker for disease recurrence and survival of HCC patients. EVs were isolated from serum samples and divided into three groups: HCC, CHB, and LC. The authors found that miR-125b levels were significantly increased in HCC-derived EVs compared to those in serum from patients with CHB or LC. Therefore, the enrichment of miR-125b in EVs can help to establish the efficacy of treatment regimens and the survival of HCC patients [120].

**miR-718.** EV-derived miR-718 can be considered a biomarker to predict HCC relapse after liver transplantation (LT). The association between the expression level of the potential miR-718 target genes in the primary HCC and post-operative prognosis was examined.

The study showed that the expression level of EV-associated miR-718 isolated from patients with HCC recurrence after LT decreased when compared to those without HCC recurrence. Additionally, in this clinical study, the downregulation of miR-718 level was directly correlated with the oncogenic homeobox protein 8 (*HOX*-B8) overexpression, resulting in tumorigenesis and tumor invasion.

The upregulation of HOXB8 expression affected the poor overall and recurrence-free survivals of HCC patients, with statistical significance [121].

**miR-122.** Another study analyzed the importance of serum EV-miRNA expression levels in HCC patients that underwent transarterial chemoembolization (TACE). Based on the relative expression of miR-122 before or after TACE, patients with a higher ratio had remarkably longer disease-specific survival than those with a lower miR-122 ratio. As a result, serum EV-derived miR-122 was confirmed as a predictive biomarker in TACE-treated HCC patients [122].

**miR-638.** The study from Shi and colleagues (2018) revealed that the serum EV derived-miR-638 not only affects the initiation, but also the progression of HCC, thus worsening the prognosis of HCC patients. The EV-derived miR-638 was downregulated in serum samples from patients with HCC compared to healthy donors. Levels of serum EV derived-miR-638 are decreased in HCC patients with larger tumor size (>5 cm) or at later TNM stage (III/IV), suggesting that the downregulation of mir-638 predicts poor prognosis in HCC patients [123].

### 4.2. Other EV-Derived RNA Molecules as Biomarkers for HCC

Recently, in a clinical study that included a large cohort comprising 159 healthy individuals, 150 patients with five cancer types, and 43 patients with other diseases, more than 10,000 EV-RNA were analyzed from human plasma.

mRNAs represented most of the total mapped reads, and the data also showed that short RNAs and circRNAs were enriched in EVs. It was analyzed whether certain EV-derived RNA could diagnose a specific tumor type, such as HCC.

Finally, eight EV-RNA molecules resulted as biomarkers for HCC diagnosis with high diagnostic efficiency [124].

Different EV-associated lncRNAs, including lncRNA-HEIH, LINC02394, LINC0635, LINC00161, and JPX, were considered diagnostic biomarkers for HCC [125,126,127,128]. LINC00853 was the last lncRNA found as a possible diagnostic biomarker of both all-stage HCC and early HCC [129].

However, the diagnostic performance of EV-derived lncRNAs in HCC still suffers the limitation of analyzing small sample sizes with a still unsatisfactory diagnostic efficiency for early-stage HCC.

In Table 1, the clinical significance of EV-derived RNAs in HCC is summarized.

**Table 1 genes-12-00902-t001:** Extracellular Vesicle-derived RNA molecules are detected as biomarkers for early diagnosis and prognosis of hepatocellular carcinoma (HCC).

Clinical Significance	EV-Derived RNA	Expression Level	Patients	Source	EV Isolation Method	Clinical Relevance	Ref.
**Detection and** **Diagnosis**	miR-21	Upregulated	30 HCC 30 CHB 30 HV	Serum	Total Exosome Isolation Kit (Invitrogen, Carlsbad, CA, USA).	Discrimination between HCC and CHB or LC	[78]
	miR-93	Upregulated	85 HCC 23 HV	Serum	Total Exosome Isolation Kit (Thermo Fisher Scientific, Waltham, MA, USA)	Biomarker for both diagnosis and prognosis in HCC.	[112]
	miR-224	Upregulated	89 HCC 50 HV	Serum	Total Exosome Isolation Kit (Thermo Fisher Scientific)	Biomarker of diagnosis and prognosis of HCC patient	[117]
	miR-718	Downregulated	59 HCC	Serum	Differential Centrifugation with a Final Ultracentrifugation Step	Predicting biomarker for recurrence after LT	[121]
	miR-18a miR-221 miR-222 miR-224 miR-101 miR-106b miR-122 miR-195	Upregulated Downregulated	20 HCC 20 LC 20 CHB	Serum	ExoQuick Exosome Precipitation Solution (System Biosciences, Mountain View, CA, USA)	Discrimination between HCC and CHB or LC	[115]
	miR-10b-5p miR-18a-5p miR-215-5p miR-940	Upregulated	90 HCC 60 CLD 28 HV	Serum	Differential Centrifugation with a Final Ultracentrifugation Step	miR-10b-5p as a diagnostic biomarker for early stage HCC	[118]
	miRNA-26a miRNA-29c miRNA-21	Downregulated	72 HCC 72 LC 72 HBV	Serum	ExoQuick Exosome Precipitation Solution (System Biosciences, Mountain View, CA, USA)	Diagnostic biomarkers for patients with HCC	[130]
	miR-122 miR-148a miR-1246	Upregulated	5 HCC 5 LC	Serum	8% Polyethylene glycol (PEG) 6000 (Sigma-Aldrich, St Louis, MO, USA)	Diagnostic biomarker for patients with HCC	[131]
	lncRNA-HEIH	Upregulated	35 CHC 22 HCV 10 HCC	Serum	Total Exosome Isolation reagent (GS0301; Guangzhou 141 Geneseed Biotech Co., Guangzhou, China) with a final centrifugation passage	Biomarker in the HCV-related hepatocellular carcinoma	[128]
	LINC00853	Upregulated	90 HCC 28 CH 35 LC 29 HV	Serum	ExoQuick Exosome Precipitation Solution (System Biosciences, Mountain View, CA, USA)	Diagnostic biomarker discriminating both all-stage HCC and early HCC	[129]
	LINC00161	Upregulated	56 HCC 56 HV 15 HCC 15 HV	SerumUrine	Total Exosome Isolation Kit (Invitrogen, USA)	Diagnostic biomarker for patients with HCC	[126]
	ENSG00000258332.1 LINC00635	Upregulated	60 HCC 85 LC 96 CHB 60 HV	Serum	Total Exosome Isolation Kit (Thermo Fisher Scientific)	Biomarker of diagnosis and prognosis of HCC patients	[127]
	Jpx	Upregulated	74 HCC 26 LC 34 CHB 72 HV	Serum	ExoQuick Exosome Precipitation Solution (System Biosciences, Mountain View, CA, USA)	Biomarkers for diagnosis of female patients with HCC	[125]
	8 EV-ncRNA (circRNA and lncRNA)	Upregulated	71 HCC (early stage, *n* = 45; advanced stage, *n* = 26) 94 HV 18 benign HCC 11 CHB 8 LC	Serum	exoRNeasy Serum/Plasma kit (Qiagen, Hilden, Germany)	Biomarkers for hepatocellular carcinoma (HCC) diagnosis	[124]
**Detection and** **Prognosis**	miR-125b	Upregulated	158 HCC 30 CHB 30 LC	Serum	ExoQuick Exosome Precipitation Solution (System Biosciences, Palo Alto, CA, USA)	Predicting biomarker for recurrence and survival	[120]
	miR-638	Downregulated	126 HCC 21 HV	Serum	Total exosome isolation kit (Invitrogen, Carlsbad, CA, USA).	Poor prognosis marker for patients with HCC	[123]
	miR-215-5p	Upregulated	90 HCC 60 CLD 28 HV	Serum	Differential Centrifugation with a Final Ultracentrifugation Step	miR-215-5p: prognostic biomarker for HCC	[118]
	miR-744	Downregulated	68 HCC 52 normal liver tissue samples	Serum	Differential Centrifugation with a Final Ultracentrifugation Step	Inhibition of Proliferation and chemoresistance	[132]
	miR-224	Upregulated	89 HCC 50 HV	Serum	Total Exosome Isolation Kit (Thermo Fisher Scientific)	Biomarker of diagnosis and prognosis of HCC patient	[117]
	miR-21 miR-10b	Upregulated	124 HCC N.A. HV	Serum	ExoQuick-TC exosome precipitation solution (System Biosciences, CA, USA)	Prognostic molecular markers and therapeutic targets for HCC.	[114]
	miR-9-3p	Downregulated	N.A. HCC N.A. HV	Serum	Differential Centrifugation with a Final Ultracentrifugation Step	Potential therapeutic target for HCC.	[116]
	circPTGR1	Upregulated	82 HCC 47 HV	Serum	ExoQuick-TC exosome precipitation solution (System Biosciences, CA, USA)	Prognostic biomarker and therapeutic target in HCC	[95]

HCC, hepatocellular carcinoma; CLD, chronic liver disease; LC, liver cirrhosis; CHB, chronic hepatitis B; HBV, hepatitis B virus; HCV, hepatitis C virus; CHC, chronic hepatitis C; CH, chronic hepatitis; HV, healthy volunteers.

## 5. Conclusions 

Research of nucleic acids within EVs to identify a panel of biomarkers has the ability to provide new biological knowledge and support diagnosis and therapeutic monitoring in HCC.

Despite the exponential interest in the EV field and the recent advances in isolation/characterization methods, the challenge of acquiring EV samples with high yield and purity and standardizing RNA processing is still open. 

It is now well known that RNA molecules are stable within EVs, since the lipid bilayer conserves them from the enzymatic activity of RNases. Therefore, to evaluate EV-derived RNAs as potential biomarkers in HCC diagnosis and prognosis, blood samples are appropriate.

Further advancement in the isolation and detection of EV-derived RNAs is required, defining the different sources from which EV-derived RNAs are obtained.

A larger cohort of HCC patients with a control cohort of healthy subjects is also necessary, considering not only the evolution of HCC progression but also other risk factors such as chronic HBV or HCV infection, alcohol abuse, LC, and aflatoxin exposure.

An important step forward will be taken when tumor properties, such as tumor differentiation stage or the post-surgery tumor relapse, including the presence of microvascular invasion, are associated with the expression level of specific EV-derived RNA molecules.

Finally, for a more relevant clinical use, a pattern of biomarkers associated with EVs in the progression of HCC may be considered. Correlation panels among RNA content and proteins and lipids associated with EVs could be set up. We are increasingly convinced that EVs contain rich information; therefore, to consider only a part of their content would be reductive from a diagnostic and prognostic perspective.

## Figures and Tables

**Figure 1 genes-12-00902-f001:**
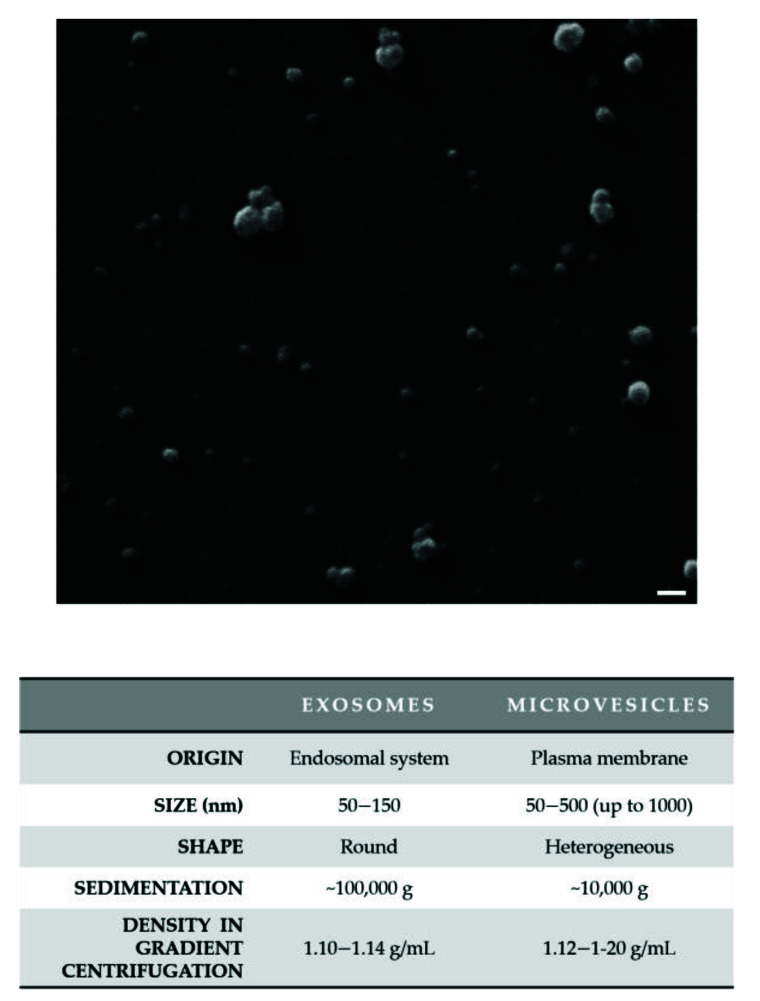
The heterogeneity of extracellular vesicles (EVs). A representative and original image of small EVs by scanning electron microscopy (SEM) is shown. EVs were isolated from the hepatocarcinoma Mahlavu cell culture medium by differential centrifugation with a final ultracentrifugation step and were characterized by electron microscopy. EVs were fixed with 2.5% glutaraldehyde in filtered PBS, sedimented onto glass coverslips and then allowed to dry at room temperature. SEM images were obtained using a SEM Zeiss EVO 40 (Zeiss; Oberkochen, Germany). EVs display their heterogeneity of size. The main features of the two main EV subtypes (Exs and MVs) are reported in the table. Scale bar: 200 nm.

**Figure 2 genes-12-00902-f002:**
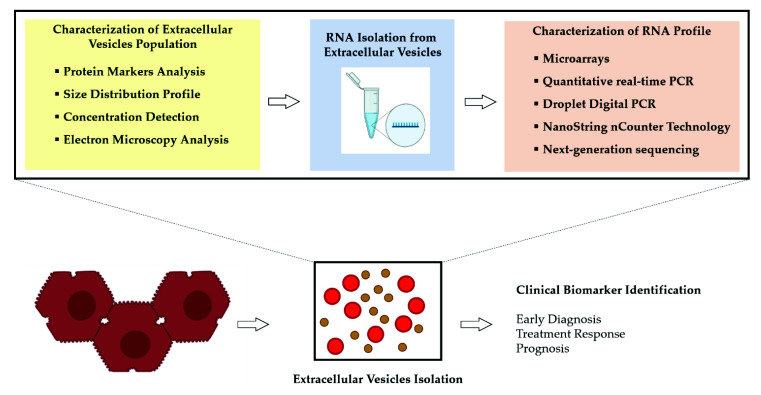
Overview of the process for the analysis of RNA molecules obtained from extracellular vesicles (EVs) released by hepatocellular carcinoma (HCC) cells. An adequate approach to isolate the EVs and the choice of assay having suitable sensitivity to detect small amounts of RNA represent the main determinants of success to discover disease-specific RNA molecules.

## Data Availability

Data sharing not applicable.

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
