# Peer review of "The Role of Extracellular Vesicles as Shuttles of RNA and Their Clinical Significance as Biomarkers in Hepatocellular Carcinoma"

_genes, 2021, doi:10.3390/genes12060902_

Round 1
Reviewer 1 Report
Dr. Costanzi and colleagues propose a review of the current status of extracellular vesicles and their clinical significance as biomarkers in hepatocellular carcinoma. The use of exosomes in the diagnostic or even treatment of HCC is still at its early stages due to several problems concerning the isolation, purification and identification of exosomes which are significant challenges to overcome. However, it is highly potential to bring breakthrough changes in clinical practice and management of HCC patient in the future.
In general, the review is well conducted and the findings are well presented, despite the fact that it is not hugely original compared to other literature reviews that are currently around in this field.
Please find below the list of the most important points to be considered while revising the article:
• My main reservation concerning of this article concerns important gaps concerning several studies which in my opinion include interesting results and are worth to be analyses by the authors since will give a broad view to the readers. I recommend the work of Dr. Shi and co-authors concerning the diagnostic and prognostic use of serum EV-miR-638 (Shi et al., J Cell Biochem. 2018) to be analyzed in the article as well the work of Yang and Zhu (Molecular Biology Reports 2014) and Hongwei Wang (Biomed Res Int, 2014). • Moreover, there is no mentioning of the latest work conducted by Xue X, Biochem Biophys Res Commun. 2018; Lu Z, J Hepatol. 2017; Tian XP, Theranostics. 2019 and Tang (Minerva Med. 2018) concerning the role of exosomal miR in HCC which I think will be of interest to the readers.
• Findings of Shikun Yang et al., (PMID: 33614242) should be analysed in the article.
• Similarly, the biological role of extracellular Vesicles, first paragraph and second; is only touched upon and important studies are ignored. Even though, I found interesting the section 5 where the authors develop the impact of microenvironment in the success of targeted therapies, this paragraph should be developed more.
• There are several paragraphs in the manuscript that lacks in-text citation; lines 252-256; lines 270; lines 323-325; lines 346-349; lines 371-373; lines 420-427; lines
• Lines: 400-406. No mention or analysis of the work of Li-ping Zhuang and Zhi-qiang Meng, (BioMed Research International, 2015)
• Lines: 416-419. Findings of Dr. Li et al., should be analyzed and presented in more details this paragraph.
Minor points
• Lines 70-77: I recommend this paragraph to be modified accordingly and to be added as figure 1 legend.
• Line 108: Please correct: can influence gene expression.
Reviewer 2 Report
This review article is interesting.
Reviewer 3 Report
The manuscript by Costanzi et al. on the role of EVs containing RNA in HCC development, and on the clinical significance of EVs as biomarkers and promising diagnostic tool in HCC offers a complete overview of EVs in HCC.
I suggest to make the corrections as reported below, re-write some sections and to include and discuss some specific points, that will complete and considerably improve the manuscript.
- Section 1. Please briefly mention the proteins and lipids EVs-associated in malignant progression, besides cav-1
- Section 2 is too descriptive in all type of RNA enclosed in EVs. I suggest to shorten this part and to briefly describe RNAs eventually by adding a Table. More, the list of miR in HCC should be shifted in section 3.
- Section 2. The existence of several procedures to isolate EVs highly secreted by tumor cells is more than an issue, and I believe that it deserves a separate section in the text. I suggest to describe the methodological procedures used to isolate EVs in HCC field, and not in general as presented in the manuscript, to provide a specific focus on this disease. Also advantages and disadvantages of each procedure should be included and discussed.
- The authors briefly mention exosomes in section 1 and later on they always mention EVs. Are there studies on the specific role of exosomes in HCC?? If yes, please include them in the text and discuss.
- Section 3. I suggest to re-write the chapter and to splite out it in two part with protumor and antitumor molecules. In this way the reading should be easier and more fluid. A discussion on biological significance and conclusions is advisable also.
- Figure 2 is incomprehensible in terms of editing and significance. Please remove TEM that is a duplicate of Fig 1, enlarge the characters and re drawn the picture under RNA profile. The figure under the square is incomprehensible. What does it mean??
Minor points:
- Lane 92. identifying ?? Please correct the term
- Lane 104. Remove has been
- Lane 274. Remove (1) and (2)
Round 2
Reviewer 1 Report
The manuscript has been sufficiently improved and therefore it is suitable for publication.
Author Response
Dear Reviewer,
Thank you for your valuable comment and work on improving our review.
Reviewer 3 Report
The authors considerably improved the manuscript that is now suitable for publication in GENES
Author Response

(The authors gave the same response as above.)
